# Dense Embeddings from Self-Supervision and Foundation Models Improve Cell Linking Performance

**Constantin Dalinghaus**[1] ⓘ       CONSTANTIN.DALINGHAUS@UNI-GOETTINGEN.DE

**Anwai Archit**[1] ⓘ       ANWAI.ARCHIT@UNI-GOETTINGEN.DE

**Constantin Pape**[1,2,3] ⓘ       CONSTANTIN.PAPE@INFORMATIK.UNI-GOETTINGEN.DE

[1] *Georg-August-University Göttingen, Institute of Computer Science*

[2] *CAIMed - Lower Saxony Center for AI & Causal Methods in Medicine, Göttingen*

[3] *Cluster of Excellence Multiscale Bioimaging (MBExC), Georg-August-University Göttingen*

## Abstract

Cell tracking is an important task in microscopy, enabling the study of cell population dynamics. The state-of-the-art uses tracking-by-detection and has substantially improved due to advances in cell segmentation and transformer-based cell linking. How can these systems be further improved once cell segmentation performance plateaus? A promising avenue is to enhance the shallow features used in cell linking with learned features. We investigate this approach by integrating learned features from self-supervised learning and foundation models within Trackastra, showing improved performance on two tracking datasets.

**Keywords:** cell tracking, microscopy, self-supervision features, foundation models

## 1. Introduction

Cell tracking is a specialized form of multiple object tracking (MOT) that focuses on following living cells in microscopy images. The field has gravitated towards methods based on the tracking-by-detection paradigm (Maška et al., 2014; Ulman et al., 2017; Maška et al., 2023), which separates the problem into cell detection (or segmentation) and cell linking. These methods have benefited from recent improvements in cell segmentation thanks to the adoption of foundation models (Archit et al., 2025; Pachitariu et al., 2025) and transformer-based architectures (Gallusser and Weigert, 2024; O'Connor and Dunlop, 2025; Zhang et al., 2025) for learned cell linking, relying mainly on morphological cell features.

This development raises the question: Once gains from improved cell segmentation are exhausted, where will future performance gains come from? While appealing from a conceptual point of view, end-to-end learning is not practical due to the lack of sufficient annotated cell tracking data. Hence, an obvious approach is to replace the morphological features used in the tracking transformer with learned features. This could be achieved through re-identification (Re-ID) features, as in general MOT (Wojke et al., 2017), and has already been done for specific cell tracking tasks (Zhou et al., 2025; Panteli et al., 2020; Ben-Haim and Raviv, 2022). However, training generalizable Re-ID features for cell tracking is also challenging due to the lack of annotated data. Instead, we investigate features from self-supervised learning (SSL) (Xie et al., 2023) and domain-specific foundation models (Archit et al., 2025). Specifically, we integrate these features within Trackastra (Gallusser and Weigert, 2024), the state-of-the-art cell linking method, and evaluate the results in two cell tracking datasets.

## 2. Methods

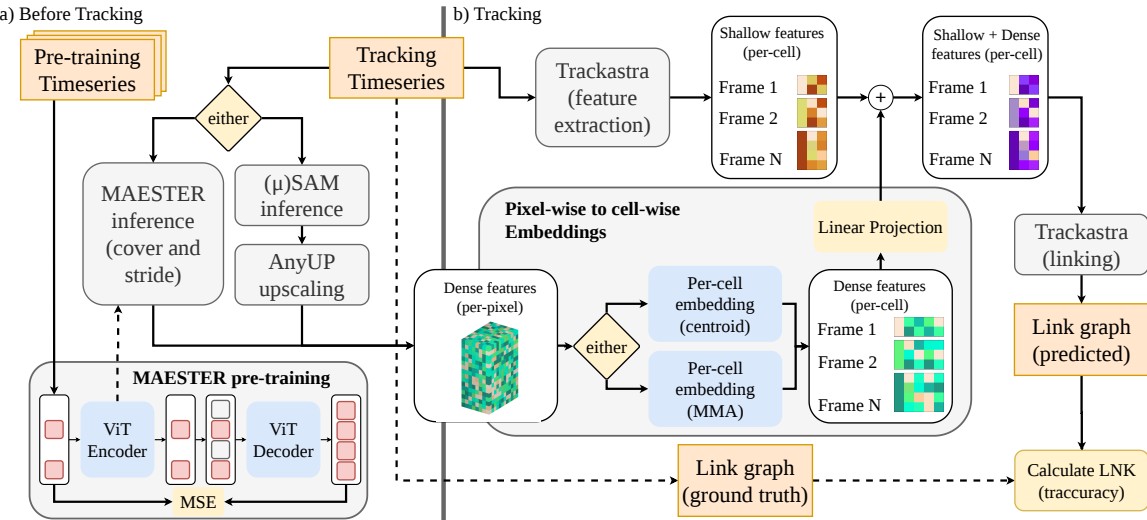

Figure 1: Pre-training process for creating pixel-wise embeddings with MAESTER / $\mu$SAM (left); pixel-wise embeddings are aggregated and fused into Trackastra (right).

### 2.1. Integrating learned features in Trackastra

We use MAESTER (Xie et al., 2023), a masked autoencoder adapted to microscopy, for embeddings from SSL. See App. A for details. We also use embeddings from foundation models: "vit_b" from (Kirillov et al., 2023) (trained on natural images) and "vit_b_lm" from $\mu$SAM (Archit et al., 2025) (trained on microscopy images). The embeddings produced by $(\mu)$SAM are of fixed spatial size (64, 64). We rescale them to match the resolution of the images with AnyUp (Wimmer et al., 2026).

We then derive a feature vector for each ground-truth cell mask from the embeddings. For this, we evaluate two approaches: (A) Select the embedding at the centroid of the cell bounding box, and (B) calculate the mean vector over the cell's mask (MMA). To integrate the feature vectors into Trackastra, we add a linear projection of size (192, 7) for MAESTER / (256, 7) for $(\mu)$SAM that maps the embedding dimension to Trackastra's feature dimension. The features are fused via addition. Except for this change, the architecture and training of Trackastra remain unchanged. See Fig. 1 for details.

### 2.2. Metrics and datasets

We use the linking accuracy measure (LNK) for evaluation. LNK was introduced in (Maška et al., 2023) as a variant of AOGM-A (Matula et al., 2015) that addresses the problems of arbitrary scale and trivial solutions. To account for statistical variability during training, we run our experiment 8 times independently and report the mean and variance. We use the DIC-C2DH-HeLa and Fluo-N2DL-HeLa datasets from (Maška et al., 2014) for evaluation. Each dataset contains two time-series with tracking ground-truth, one for training, one for evaluation. See App. B for details about data and the experiment setup.

## 3. Results

Table 1: Performance of default Trackastra vs. our version with features from $(\mu)$SAM / MAESTER. Variance is derived from 8 training runs with different seeds.

| Dataset | Method / Features | LNK (Default) | LNK (MMA) | LNK (Centroid) |
|---|---|---|---|---|
| Fluo-N2DL-HeLa | Trackastra (re-trained) | .9922 ± 3.19e-07 | - | - |
| Fluo-N2DL-HeLa | Trackastra (pretrained) | .9904 | - | - |
| Fluo-N2DL-HeLa | SAM (vit_b) | - | .9923 ± 2.4e-07 | .9924 ± 4.6e-07 |
| Fluo-N2DL-HeLa | $\mu$SAM (vit_b_lm) | - | .9930 ± 2.1e-07 | .9930 ± 2.0e-07 |
| Fluo-N2DL-HeLa | MAESTER | - | .9935 ± 4.2e-07 | **.9944** ± 4.2e-07 |
| DIC-C2DH-HeLa | Trackastra (re-trained) | .9859 ± 1.78e-05 | - | - |
| DIC-C2DH-HeLa | Trackastra (pretrained) | .9815 | - | - |
| DIC-C2DH-HeLa | SAM (vit_b) | - | .9890 ± 5.9e-06 | .9884 ± 1.5e-05 |
| DIC-C2DH-HeLa | $\mu$SAM (vit_b_lm) | - | **.9898** ± 1.8e-06 | .9869 ± 2.6e-06 |
| DIC-C2DH-HeLa | MAESTER | - | .9857 ± 4.1e-06 | .9805 ± 1.7e-05 |

The comparison of default Trackastra (trained only on the respective training set, re-trained; the publicly available model, pretrained) with our versions that use learned features is presented in Tab. 1. On Fluo-N2DL-HeLa, our method performs better than default Trackastra in all settings, with MAESTER features and centroid aggregation performing best. On DIC-C2DH-HeLa, the features derived from $\mu$SAM (vit_b_lm) perform best, followed by features from SAM (vit_b). They perform clearly better compared to default Trackastra; unlike the version trained with MAESTER features. Comparing MMA and centroid aggregation shows that MMA is beneficial for $(\mu)$SAM-derived features, whereas the trend is less clear for MAESTER.

We also study the effect of the amount of data and compute on self-supervised pre-training as well as their transferability to other data, see App. C. Briefly, we find that advantages from increasing data and compute are limited, and that features transfer well across image data from the same source domain.

## 4. Discussion

We found that linking performance improves when integrating learned features into cell linking systems, showcasing how cell tracking can be advanced beyond segmentation improvements. Furthermore, our experiments showed the utility of pre-trained foundation models in deriving these features, namely SAM and $\mu$SAM, and showed the advantage of domain-specific features from $\mu$SAM over SAM. Comparing foundation model features and specifically trained SSL features showed inconsistent results; one of the datasets showed advantages of SSL features, but they performed clearly worse, not even exceeding the baseline, in the other. Given these observations, foundation model features are currently preferable, as they are also much easier to use in practice.

Future work could explore more powerful feature representations that also take the time axis into account, such as features from (microscopy-specific) video foundation models, building for example on V-JEPA models (Bardes et al., 2024).

## Acknowledgments

We acknowledge the use of the Mitocheck database while conducting this study. The work of Anwai Archit was funded by the Deutsche Forschungsgemeinschaft (DFG, German Research Foundation) - PA 4341/2-1. Constantin Pape is supported by the German Research Foundation (Deutsche Forschungsgemeinschaft, DFG) under Germany's Excellence Strategy - EXC 2067/1-390729940. This work is supported by the Ministry of Science and Culture of Lower Saxony through funds from the program zukunft.niedersachsen of the Volkswagen Foundation for the 'CAIMed – Lower Saxony Center for Artificial Intelligence and Causal Methods in Medicine' project (grant no. ZN4257). It was also supported by the Google Research Scholarship "Vision Foundation Models for Bioimage Segmentation". We gratefully acknowledge the computing time granted by the Resource Allocation Board and provided on the supercomputer Emmy at NHR@Göttingen as part of the NHR infrastructure, under the projects nim00007 and nim00020.

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

## Appendix A. MAESTER Set-up

We base our method for learning self-supervised pixel-wise embeddings on work by (Xie et al., 2023), which in turn builds on the training principle and architecture introduced by (He et al., 2022).

(Xie et al., 2023)'s main architectural contribution is the 'cover-and-stride' inference technique, which uses a MAE-encoder to infer pixel-wise embeddings for arbitrary two dimensional slices of a volume. 'Cover-and-stride' works on random crops from the two dimensional slices, which (Xie et al., 2023) call FOVs. It works by assigning the embedding value produced by the MAE-encoder for each patch to the pixel at the center of the respective patch. To avoid edge bias, (Xie et al., 2023) apply the patch-to-pixel assignment only for patches close to the center of the image crop. This iterative method of doing inference results in a two-layered striding process ('cover' / 'stride'). Specifically, the inference process requires passing patches through the trained MAE encoder for a specific FOV, then move the FOV by one pixel, until the pixels in the center region of the FOV are fully covered ('cover'). To fully process an entire image, the FOV has then to be moved to the next location to fill the adjacent region with pixel-wise embeddings ('stride'). The 'cover-and-stride' inference technique can be described as resource-intensive.

The MAE pre-training technique as introduced by (He et al., 2022) is originally designed for processing two dimensional images. To adapt this method for a three dimensional volumetric setting, (Xie et al., 2023) introduce a mechanism that randomly selects two of the three axes from their three dimensional volumetric data when a training example is requested from the DataLoader[1]. Along these randomly selected dimensions, a two dimensional slice is taken, from which a random FOV is then selected for processing. This way, in (Xie et al., 2023), MAE handles three dimensional volumetric data, despite being designed to process two dimensional data. In our method, we represent a 2D+Time dataset as a three dimensional volume. To do so, we reuse one of the axes as the temporal axis. To avoid slicing along the temporal axis, we remove the random selection of the slice axes from the MAESTER pre-training code[2].

We note that the patch size is required to be an uneven integer, for 'cover-and-stride' inference to work, a restriction that the original MAE implementation does not have. For our method, we keep the hyperparameter selection identical to the ones used in (Xie et al., 2023). The specific hyperparameters used during our pre-training process are documented in Tab. 2.

We reuse the code made available by (Xie et al., 2023) for pre-training and 'cover-and-stride' inference. To make MAESTER functional in our environment, and especially for multi-GPU training, we modify the original implementation. These alterations have minor implications for how the loss value is recorded and change the behavior of the pseudo-random

---

1. https://github.com/bowang-lab/MAESTER/blob/e19e020d02520777ff0996f21ee5cccb98f645ef/MAESTER/dataset.py#L90
2. https://github.com/MIDL26-Short-Tracking/MAESTER_fix/commit/a001ec3fca50da46bba981dafb80c3c866428f78#diff-da4bee931db24c0b6f470919a2735218690afde9730df4dfb60100fb99da69f9

number generator. The code used in our method is available on GitHub[3], alterations to the original implementation can be followed in the commit history[4].

**Loss value recording**  Our modified code for MAESTER is designed to run on 4 x NVIDIA A100 GPUs using data parallel. Our implementation records the loss value as calculated on the first GPU (gpu0). Loss values as calculated on (gpu1, gpu2, gpu3) are not recorded. Exact behavior can be studied in lines 89 and 99 of train.py from the MAESTER_fix repository[5].

**Pseudorandom number generator**  In the original implementation of MAESTER, in the file "train.py", a call to (set_deterministic) is used to set a fixed seed value for the pseudorandom number generator. This is presumably done to ensure reproducibility. The DataLoader is set up in a way that re-calls (set_deterministic) (through the (seed_worker) function) before every train epoch, resulting in identical random crops for every epoch. It is unclear whether this is intended behavior for MAESTER. Our method aims to make use of the pre-training data as thoroughly as possible. For this reason, the function (seed_worker) is disabled in our modified version of the implementation, resulting in random crops that are truly pseudo-random[6].

Table 2: Pre-training parameters used in our method are consistent with the parameters used in (Xie et al., 2023) (default configuration file). Note that MAESTER uses a mask ratio of 0.5 instead of the higher ratios (He et al., 2022) determined to be optimal for natural images.

| Hyperparameter | Fluo-N2DL-HeLa | DIC-C2DH-HeLa |
|---|---|---|
| img_size | 80 | 80 |
| patch_size | 5 | 5 |
| embed_dim | 192 | 192 |
| depth | 14 | 14 |
| num_heads | 1 | 1 |
| decoder_embed_dim | 128 | 128 |
| decoder_depth | 7 | 7 |
| decoder_num_heads | 8 | 8 |
| mlp_ratio | 2.0 | 2.0 |
| mask_ratio | 0.5 | 0.5 |
| pos_encode_w | 0.08 | 0.08 |
| central_patch | 4 | 4 |
| batch_size | 32 | 32 |
| adamw_lr | 6.0e-5 | 6.0e-5 |
| adamw_reg_w | 1.0e-5 | 1.0e-5 |

---

3. https://github.com/MIDL26-Short-Tracking/MAESTER_fix

4. https://github.com/bowang-lab/MAESTER/compare/main...MIDL26-Short-Tracking:MAESTER_fix:main

5. https://github.com/MIDL26-Short-Tracking/MAESTER_fix/blob/main/MAESTER/train.py#L98

6. https://github.com/MIDL26-Short-Tracking/MAESTER_fix/blob/quadgpu/MAESTER/train.py#L72

## Appendix B. Experimental Setup

For evaluation of the tracking system, we use the DIC-C2DH-HeLa and Fluo-N2DL-HeLa datasets from (Maška et al., 2014). The microscopy images in Fluo-N2DL-HeLa are captured using an Olympus IX81 inverted epifluorescence microscope with a Plan 10x/0.4 lens at a pixel size of 0.645 x 0.645 microns. Images are captured on a 30-minute time interval[7] (Neumann et al., 2010). DIC-C2DH-HeLa[8] contains high contrast cell imaging data at high density. The microscopy images in DIC-C2DH-HeLa are captured using a Zeiss LSM 510 Meta microscope and a Plan-Apochromat 63x/1.4 (oil) lens. Pixel size was 0.19 x 0.19 (in microns). The time step between consecutive frames is 10 minutes[9]. Both datasets contain a train set with two labeled sequences. We do not use the test-set, as it has no labels. The sequences of DIC-C2DH-HeLa contain 84 images of size 512 x 512 each, the sequences of Fluo-N2DL-HeLa contain 92 images of size 1100 x 700. We use the '02'-sequence for training and the '01'-sequence for evaluation, adapting the approach from (Gallusser and Weigert, 2024).

As mentioned in the method section, training of the Trackastra linking system is always run 8 times independently of each other. We then report the mean of the LNK values, alongside the estimated variance. We do this in an attempt to quantify the robustness of our results.

Our linking experiments use four different configurations of Trackastra. Trackastra (re-trained) trains a linking system without added embeddings, Trackastra (pre-trained) only evaluates the linking system. Against these baselines, we compare two aggregation strategies for the pixel-wise embeddings (MMA / Centroid). The code for the experiments is organized via different branches on GitHub:

**Trackastra (re-trained)** is a Trackastra system without any modifications, based on Trackastra 0.4.1. We use this branch to re-train a Trackastra model without added embeddings, as originally intended by (Gallusser and Weigert, 2024). Code can be found on the counterfactual branch of our modified Trackastra repository [10]. **Trackastra (pre-trained)** only evaluates tracking results using the generalist Trackastra weights from Trackastra 0.4.1. This setting uses the baseline Trackastra repository with the "eval_pretrained.py" script for evaluation. **Trackastra (MMA)** uses the Mean-Mask-Aggregation strategy to convert pixel-wise embeddings to cell-wise embeddings. Our modifications for this Trackastra system are based on Trackastra 0.4.1. Code can be found on the "linear_mean_mask_aggregation" branch[11]. **Trackastra (Centroid)** uses the embedding at the centroid location of each cell mask to convert pixel-wise embeddings into cell-wise embeddings. Code for this Trackastra system can be found on the fusion / linear branch [12].

---

7. Technical information from https://celltrackingchallenge.net/2d-datasets/, "inverted epifluorescence" from (Ulman et al., 2017)

8. Provided to the celltrackingchallenge by Dr. G. van Cappellen. Erasmus Medical Center, Rotterdam, The Netherlands https://celltrackingchallenge.net/2d-datasets/

9. https://celltrackingchallenge.net/2d-datasets/

10. https://github.com/MIDL26-Short-Tracking/trackastra_y/tree/counterfactual

11. https://github.com/MIDL26-Short-Tracking/trackastra_y/tree/fusion/linear_mean_mask_aggregation

12. https://github.com/MIDL26-Short-Tracking/trackastra_y/tree/fusion/linear

**HPC training setup**  The tracking system is trained on an HPC system using a single NVIDIA A100 GPU. The slurm configuration files for the reproduction of results can be found on GitHub[13]. After every training epoch, a full model checkpoint is persisted to disk.

**Evaluation setup**  For runs that involve model training (all except Trackastra pre-trained), inference is run using the eval_combined.py script[14]. The script iterates over all model checkpoints that were saved during training. For each checkpoint, the predicted lineage graph is computed, which is then compared to the ground truth linking graph, as provided by the CTC. For calculation of the LNK metric, we use the implementation provided by the traccuracy python library (https://github.com/live-image-tracking-tools/traccuracy). Inference is run on CPU. Slurm config files for reproducing results are made available on GitHub[15].

---

13. https://github.com/MIDL26-Short-Tracking/trackastra_y
14. https://github.com/MIDL26-Short-Tracking/trackastra_y/blob/main/scripts/eval_combined.py
15. https://github.com/MIDL26-Short-Tracking/trackastra_y/blob/main/scripts/array.sbatch

## Appendix C. Additional studies on self-supervised features

We are interested in a possible relationship between the amount of data available during pre-training and the resulting linking performance, as well as the relationship between the amount of compute applied during pre-training and the resulting linking performance. For this, we investigate how a single dataset and aggregation strategy behave under a varied amount of pre-training compute and pre-train data. For the following experiments we select the Fluo-N2DL-HeLa dataset under a centroid-aggregation strategy.

### C.1. Scaling pre-training compute

First, we investigate the relationship between the amount of compute applied during pre-training and the resulting tracking performance in LNK. For this, we extend our method to save model checkpoints continuously during MAESTER pre-training. From these checkpoints, we select a subset of checkpoints roughly every 2000 steps. We select additional checkpoints early during pre-training to increase the resolution of our measurements. In total, we select 15 MAESTER checkpoints, trained for the following number of steps: (49, 149, 449, 1399, 2399, 4399, 6399, 8399, 10399, 12399, 14399, 16399, 16599, 17599, 18249). Please note that despite our best efforts, evaluation for some of these runs failed due to computational issues. See all available data on GitHub [16].

For every MAESTER checkpoint selected, we infer the corresponding embedding volume and use these embeddings to train and evaluate a Trackastra model 8 times, as per our main method. The idea is that since later embedding volumes are derived using later pre-training checkpoints, later embedding volumes should also contain pixel-wise embeddings with more nuanced representations. Later embeddings should hypothetically be more helpful to the linking system.

We observe that the amount of compute applied during pre-training has a strong positive influence on mean linking performance early during pre-training, but improvements to linking performance wear off after our measurement for pre-training step 1399. Until pre-training step 4399, mean linking performance stagnates. Following pre-training step 4399, mean linking performance appears to trend downward slightly (Figure 2).

### C.2. Transferability of pre-training

Next, we were interested to see if our method also improves tracking performance when the image sequence to be tracked is not part of the pre-training dataset. Since the Fluo-N2DL-HeLa dataset (as available on the CTC website) is a labeled subset of the Mitocheck repository, for this experiment, we randomly select one additional plate of image sequences from the Mitocheck dataset (Neumann et al., 2010). For this experiment, we use plate 1, as indicated in Tab. 3. We use all 318 sequences from this plate. The sequences from Mitocheck have a resolution of 1344 x 1024, which is larger than the images in the Fluo-N2DL-HeLa tracking dataset. Our interpretation is that the images from Fluo-N2DL-HeLa are center-crops of an image sequence from the Mitocheck dataset, which was presumably done to allow for accurate human annotation along the edges of the frame.

---

16. https://github.com/MIDL26-Short-Tracking/figures/tree/main/appendix

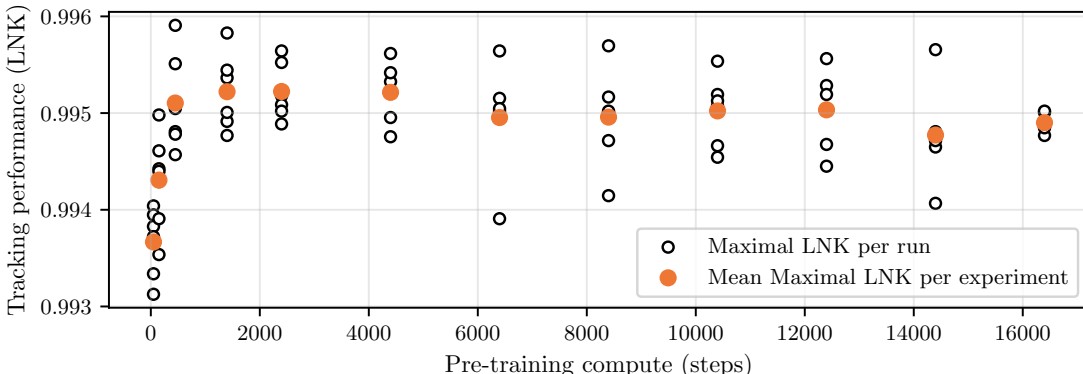

Figure 2: Mean linking performance improves when more compute is applied during pre-training; the effect weakens after pre-training step 1399; after pre-training step 4399, linking performance appears to decrease slightly

Table 3: Plates selected from the Mitocheck repository for pre-training (Neumann et al., 2010); The Fluo-N2DL-HeLa tracking dataset is a labeled subset of the Mitocheck repository; Numeric ids (1-4) are assigned by us; mitocheck_plate_ids are assigned by the Mitocheck consortium

| Plate Name | Plate ID | Mitocheck Plate ID | # of Sequences |
|---|---|---|---|
| plate 1 | 1 | dfdcb304-ffe1-4844-bae3-9ee1ed4749a7 | 318 |
| plate 2 | 2 | fa92aee7-424d-4b12-8557-c957971f8400 | 384 |
| plate 3 | 3 | f2305557-54d0-41fa-acab-5b7d6abf435e | 384 |
| plate 4 | 4 | fa2d73ab-af7c-43f9-b6a9-996566867a99 | 318 |

Table 4: For studying the effect of pre-training dataset size, we combine the plates selected from the Mitocheck repository into datasets of increasing size. For plate selection, see Tab. 3.

| Dataset Name | Dataset ID | # Plates Included | # of Sequences |
|---|---|---|---|
| Mitocheck (100%) | 1 | plate1 | 318 |
| Mitocheck (200%) | 2 | plate1, plate2 | 702 |
| Mitocheck (300%) | 3 | plate1, plate2, plate3 | 1086 |
| Mitocheck (400%) | 4 | plate1, plate2, plate3, plate4 | 1404 |

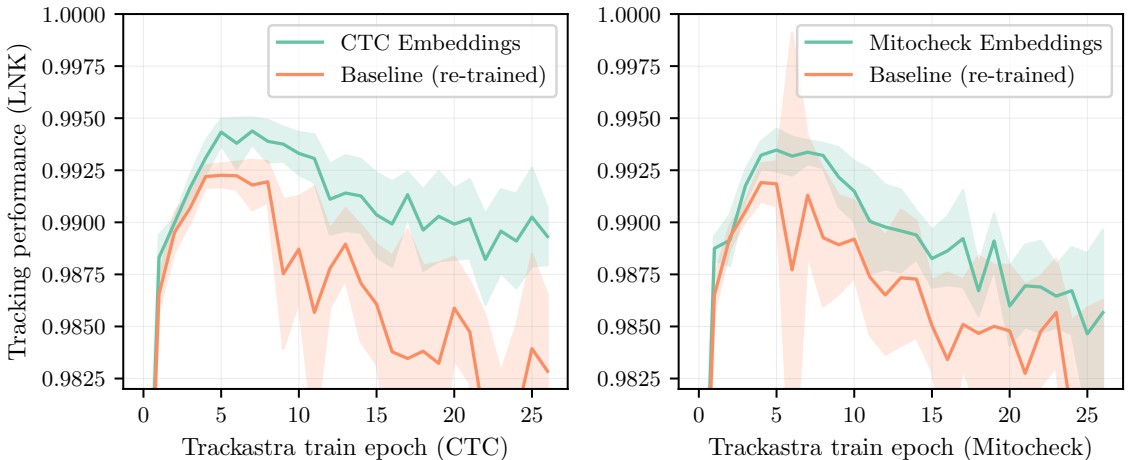

Figure 3: Embeddings from MAESTER pre-training have a positive effect on mean linking performance when pre-trained in both cases: (A) including the image sequence used for tracking (left) and (B) excluding the image sequence used for tracking (right)

Figure 3 shows cell linking performance on Fluo-N2DL-HeLa over the duration of Trackastra training. The positive influence for using embeddings from a MAESTER model pre-trained on the tracking dataset is also observed when pre-training includes different image sequences from the Mitocheck repository.

## C.3. Scaling pre-training compute and pre-training data

Motivated by the results in Section C.2, we investigate whether a relationship between pre-training dataset size and cell-linking performance can be established. For this, we extend the experiment from Section C.2 by selecting three additional image sequences from the Mitocheck dataset. From these four Mitocheck sequences, we then construct four pre-training datasets at 100%, 200%, 300% and 400% size (Tab. 3, Tab. 4). We run the experiment previously introduced in Section C.2 and compare mean linking performance across different pre-training dataset sizes.

Fig. 4 shows the mean cell linking performance for different dataset sizes, organized on the x-axis by number of steps applied during pre-training. Data points are missing, notably for early pre-training on the Mitocheck 100% dataset, and also at various points during training on the larger datasets. See all available data on GitHub [17]. The data we have observed indicates lower performance for pre-training on the Mitocheck 100% dataset, but only after the peak in linking performance is reached for pre-training on the other datasets. Our data on pre-training on the Mitocheck 200%, Mitocheck 300% and Mitocheck 400%

---

17. https://github.com/MIDL26-Short-Tracking/figures/tree/main/appendix

dataset show no clear positive effect for the influence of datasets larger than Mitocheck 100% on cell linking performance.

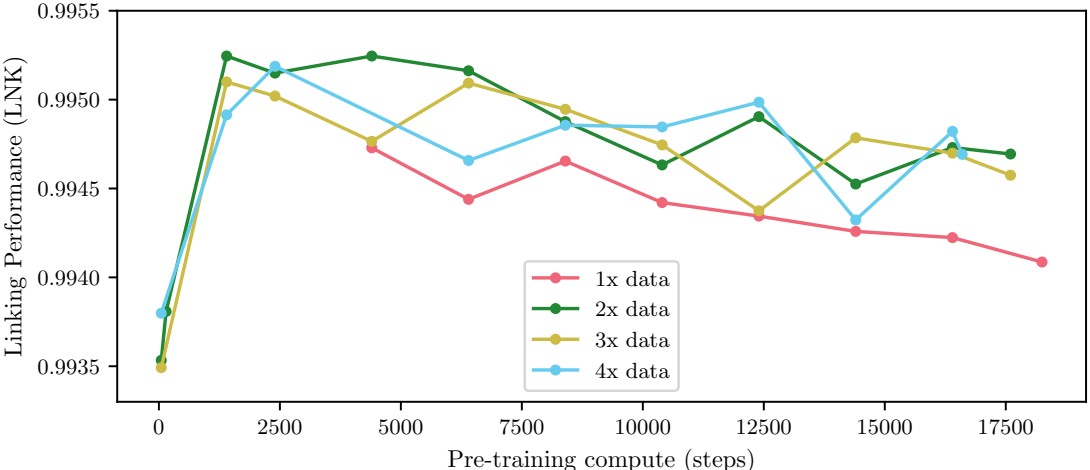

Figure 4: Showing mean cell linking performance (LNK) when the amount of pre-training compute (x-axis) and the size of the pre-training dataset is varied (color); Data for certain run configurations is not available

