# OpenReview forum: "Dense Embeddings from Self-Supervision and Foundation Models Improve Cell Linking Performance"
_MIDL.io/2026/Short_Papers — MIDL 2026 - Short Papers Poster_

### Official Review · Reviewer_srvp · 2026-05-03

**Rating:** 4
**Confidence:** 4

**Review:**

This paper presents a clear and well-motivated extension to existing cell tracking pipelines by incorporating learned dense embeddings into the linking stage, addressing an important bottleneck when segmentation performance saturates. The method is technically straightforward yet effective, and the experimental results demonstrate consistent, albeit modest, improvements across datasets. While the novelty is somewhat incremental and limited to feature integration rather than architectural innovation, the work is practically relevant and well-aligned with current trends in leveraging foundation models. Overall, the paper is clearly written, methodologically sound, and provides useful insights, though deeper analysis and broader validation would strengthen its impact.

**Summary:**

This paper investigates how to improve cell tracking performance when segmentation quality has plateaued, focusing specifically on the cell linking stage in tracking-by-detection pipelines. The authors propose augmenting the Trackastra framework with dense learned embeddings derived from self-supervised learning (MAESTER) and foundation models (SAM and µSAM), converting pixel-wise embeddings into per-cell features via centroid or mask-based aggregation. Experiments on two Cell Tracking Challenge datasets (Fluo-N2DL-HeLa and DIC-C2DH-HeLa) show consistent improvements over baseline Trackastra, particularly with foundation model features, while self-supervised features exhibit dataset-dependent behavior. The study highlights that incorporating pretrained representations can enhance linking accuracy without requiring additional annotations, offering a practical direction for advancing cell tracking beyond segmentation improvements.

**Strengths:**

1. The proposed approach is simple, modular, and easy to integrate into existing pipelines (Trackastra), increasing its practical applicability.
2. The use of both self-supervised and foundation model features provides a comprehensive comparison of representation types.
3. The findings offer actionable insights, particularly the strong performance and usability of foundation model features (µSAM).

**Weaknesses:**

1. The methodological novelty is limited, as the work mainly explores feature augmentation rather than proposing a fundamentally new model.
2. Performance gains, while consistent, are relatively small and may not be practically significant in all scenarios.
3. The analysis of why self-supervised features underperform in some cases remains shallow and could be further explored.

**Justification Of Rating:**

This paper provides a solid and practically useful contribution by demonstrating that dense learned embeddings—especially from foundation models—can improve cell linking performance with minimal modification to existing pipelines. While the novelty is incremental and the empirical gains are modest, the work is well-executed, clearly presented, and aligned with current research directions. Given the inclusive nature of the short paper track and the absence of critical methodological flaws, I recommend acceptance.

---

### Decision · Program_Chairs · 2026-05-08

Accept (Poster)